# A Suboptimal Diet Is Associated with Poorer Cognition: The NUDAD Project

**DOI:** 10.3390/nu12030703

**Published:** 2020-03-06

**Authors:** Jay L. P. Fieldhouse, Astrid S. Doorduijn, Francisca A. de Leeuw, Barbara J. H. Verhaar, Ted Koene, Linda M. P. Wesselman, Marian A.E. de van der Schueren, Marjolein Visser, Ondine van de Rest, Philip Scheltens, Maartje I. Kester, Wiesje M. van der Flier

**Affiliations:** 1Alzheimer Center Amsterdam, Department of Neurology, Amsterdam Neuroscience, Amsterdam UMC, Vrije Universiteit Amsterdam, 1081HZ Amsterdam, The Netherlands; a.doorduijn@amsterdamumc.nl (A.S.D.); f.deleeuw@amsterdamumc.nl (F.A.d.L.); b.j.verhaar@amsterdamumc.nl (B.J.H.V.); t.koene@amsterdamumc.nl (T.K.); l.wesselman@amsterdamumc.nl (L.M.P.W.); p.scheltens@amsterdamumc.nl (P.S.); mkester@flevoziekenhuis.nl (M.I.K.); wm.vdflier@amsterdamumc.nl (W.M.v.d.F.); 2Department of Nutrition and Dietetics, Amsterdam UMC, Vrije Universiteit Amsterdam, Public Health Research Institute Amsterdam, 1081HV Amsterdam, The Netherlands; marian.devanderschueren@han.nl; 3Department of Nutrition and Health, HAN University of Applied Sciences, 6525EJ Nijmegen, The Netherlands; 4Department of Health Sciences, Faculty of Science, Vrije Universiteit Amsterdam and the Amsterdam Public Health Research Institute, 1081HV Amsterdam, The Netherlands; m.visser@vu.nl; 5Division of Human Nutrition and Health, Wageningen University & Research, 6708WE Wageningen, The Netherlands; ondine.vanderest@wur.nl; 6Department of Epidemiology and Biostatistics, Amsterdam Neuroscience, Amsterdam UMC, Vrije Universiteit Amsterdam, 1081HV Amsterdam, The Netherlands

**Keywords:** nutrition, food intake, neuropsychological functioning, mild cognitive impairment, dementia

## Abstract

Nutrition is one of the modifiable risk factors for cognitive decline and Alzheimer’s disease (AD) dementia, and is therefore highly relevant in the context of prevention. However, knowledge of dietary quality in clinical populations on the spectrum of AD dementia is lacking, therefore we studied the association between dietary quality and cognitive impairment in Alzheimer’s disease (AD) dementia, mild cognitive impairment (MCI) and controls. We included 357 participants from the NUDAD project (134 AD dementia, 90 MCI, 133 controls). We assessed adherence to dietary guidelines (components: vegetables, fruit, fibers, fish, saturated fat, trans-fat, salt, and alcohol), and cognitive performance (domains: memory, language, visuospatial functioning, attention, and executive functioning). In the total population, linear regression analyses showed a lower vegetable intake is associated with poorer global cognition, visuospatial functioning, attention and executive functioning. In AD dementia, lower total adherence to dietary guidelines and higher alcohol intake were associated with poorer memory, a lower vegetable intake with poorer global cognition and executive functioning, and a higher trans-fat intake with poorer executive functioning. In conclusion, a suboptimal diet is associated with more severely impaired cognition—this association is mostly attributable to a lower vegetable intake and is most pronounced in AD dementia.

## 1. Introduction

The prevalence of Alzheimer’s disease (AD) dementia is increasing, and it is one of the largest health care challenges of our time [1]. Nutrition is one of the modifiable risk factors for cognitive decline and AD dementia [2], and is therefore highly relevant in the context of prevention. In cognitively normal populations, food groups including vegetables, fruit, or fish [3,4,5,6,7,8], and healthy dietary patterns such as the Mediterranean diet, dietary approaches to stop hypertension (DASH), or Mediterranean-DASH intervention for neurodegenerative delay (MIND) [9,10,11,12,13], have all been associated with less cognitive decline or reduced risk for mild cognitive impairment (MCI) or AD dementia. However, knowledge of dietary quality in clinical populations on the spectrum of AD dementia is lacking, while these patients may form a target group for interventions to prevent further cognitive decline. A recent study found that better adherence to the Mediterranean and MIND diets was associated with better global cognition in cognitive healthy adults, but not in MCI or AD dementia patients [14]. Most previous studies used a global cognitive screener, such as the Mini Mental State Examination (MMSE) [15], while in patients across a cognitive continuum, an extensive neuropsychological assessment provides a more sensitive outcome.

Therefore, we assessed the adherence to the Dutch Healthy Dietary guidelines on eight dietary components, and investigated associations between dietary quality and performance on the cognitive domains of memory, language, visuospatial functioning, attention, executive functioning, and global cognition in patients with AD dementia, MCI, and controls.

## 2. Materials and Methods

### 2.1. Study Population

The ‘Nutrition the Unrecognized Determinant for Alzheimer’s Disease’ (NUDAD) project prospectively studies nutritional determinants in AD dementia and pre-dementia stages [16]. The NUDAD project is a nested study within the Amsterdam Dementia Cohort [17], with additional nutritional measurements. Participants were included between September 2015 and August 2017 and were eligible if they had a clinical diagnosis of AD dementia, MCI, or subjective cognitive decline (SCD), and if the MMSE score was >16. For the current study, we investigated cross-sectional baseline data of all NUDAD participants with available neuropsychological and dietary quality data. All participants underwent a standardized dementia screening, including extensive neuropsychological assessment, neurological examination, and laboratory tests [17]. Clinical diagnosis of MCI and AD dementia was established by consensus in a multidisciplinary meeting according to the National Institute on Aging-Alzheimer’s Association criteria [18,19]. MCI is an umbrella term characterizing cognitive impairment beyond what is expected for age and education, but not affecting activities of daily living and not as severe to fulfill the criteria for dementia [20]. As a control group, we used participants with SCD who presented with memory complaints, but performed normally on all clinical examinations, i.e., criteria for MCI, dementia, or psychiatric diagnosis were not fulfilled [17]. In total, 357 participants were included in the current manuscript, of which 134 were AD dementia patients, 90 MCI patients, and 133 controls. The study was approved by the Ethics Committee of the Amsterdam UMC (2015.457). All participants provided written informed consent to use their clinical data for research purposes. Age, sex, level of education, body mass index (BMI, in kg/m^2^), and MMSE score of all participants were assessed. Level of education was classified using the Verhage system ranging from 1 (no or little education) to 7 (highest academic degree) [21].

### 2.2. Dietary Quality

The validated Dutch Healthy Diet-Food Frequency Questionnaire (DHD-FFQ) was used to assess averaged daily dietary quality in the previous month [22,23]. This questionnaire comprised 34 items, covering eight dietary components: vegetables, fruit, fibers, fish, saturated fat, trans fat, salt, and alcohol. Frequency, ranging from ‘never’ to ‘every day’, and portion size in household measures were asked. Each component score ranged from 0 to 10, with a total score ranging from 0 to 80 [23]. A higher score indicated a better adherence to the Dutch guidelines for a healthy diet [24]. Lower scores on the components vegetables, fruit, fibers, and fish indicated a lower intake, while lower scores on the components saturated fat, trans fat, salt, and alcohol indicated a higher intake. When a patient was not able to fill in the questionnaire, the DHD-FFQ was completed by a caregiver.

### 2.3. Neuropsychological Assessment

Cognitive performance was assessed using a standardized neuropsychological test battery. A test battery of 12 tests was used to measure five cognitive domains: memory, language, visuospatial functioning, attention, and executive functioning. The domain memory included the total recall on Visual Association Test (VAT), and the total immediate and delayed recall of the Dutch version of the Rey Auditory Verbal Learning Task [25,26]. The domain language included the category fluency (animal naming) and the naming condition of the VAT [26,27]. The domain visuospatial functioning included the three subtests of the visual object and space perception battery: dot counting, fragmented letters, and number location [28]. The domain attention included the trail making test (TMT) part A, the forward condition of digit span, and the word and color subtests of the Stroop test [29,30,31]. The domain executive functioning included the frontal assessment battery, the backward condition of digit span, the color-word subtest of the Stroop test and letter fluency [30,31,32,33]. Raw test scores were converted into z-scores using the mean and standard deviation (SD) of our study population. The test scores of pace dependent tests (TMT A and Stroop subtests) were not normally distributed, and therefore log-transformed and inverted to ensure lower scores indicating poorer cognitive performance. Global cognitive performance was estimated by averaging the z-scores of all tests. Domain z-scores were calculated by averaging the z-scores of all tests within the concerned domain, if at least two tests were available. Missing values comprised 2.5% in the domain memory, 3.4% in language, 9.0% in visuospatial functioning, 2.0% in attention, and 4.5% in executive functioning.

### 2.4. Data Analyses

Between-group differences in demographics were tested using analysis of variance (ANOVA) with post-hoc Bonferroni adjusted *t*-tests or Chi-Square-tests when appropriate. Between-group differences in dietary quality and cognitive performance were tested using ANOVA adjusted for age, sex, education, and BMI, with a post-hoc Bonferroni adjusted *t*-test. Associations between dietary scores (total and components, independent variables) and cognitive performance (global and domains, dependent variables in separate models) were assessed using linear regression analyses, adjusted for age, sex, education, and BMI. In the first model, we analyzed the total study population. In the second model, we estimated the effect sizes for each group by including dummy variables and interaction terms (dummy diagnosis × dietary component) in the model. Standardized betas with 99% confidence interval (CI) are presented. Significance was set at *p* value < 0.01 for all analyses. All statistical analyses were performed with SPSS version 22 (released 2013, IBM SPSS Statistics for Windows, Armonk, NY, USA). Plots were created with RStudio 3.4.2 (Windows) using the forestplot package [34].

## 3. Results

The total sample was on average 65 ± 8.3 years of age, consisting of 165 (46%) females, and had a mean BMI of 26 ± 4.1 (Table 1). AD dementia and MCI patients were older than controls, and AD dementia patients had a lower level of education and a lower BMI than controls. Regarding dietary quality, the total sample had a mean total dietary score of 54 (cut-off adequate adherence = 53 [20]). On average, participants adhered best to the guidelines for alcohol intake and less to the guidelines for saturated fat intake. The groups did not differ in total dietary score or in dietary component scores. By definition, cognitive performance was worst in AD dementia, with MCI in between AD dementia and controls.

In the total study population, linear regression analyses adjusted for age, sex, education, and BMI showed no association between the total dietary score and cognitive performance (Table 2). When we evaluated specific food groups, we found that a lower vegetable score (i.e., lower intake) was associated with poorer global cognition (sβ 0.18 (0.01; 0.07)), and poorer performance in the domains visuospatial functioning (sβ 0.15 (0.00; 0.08)), attention (sβ 0.17 (0.01; 0.09)), and executive functioning (sβ 0.13 (0.00; 0.08)). No associations were found between other dietary component scores and cognitive performance.

Subsequently, we estimated effect sizes by group and found that associations between dietary quality and cognitive performance were largely attributable to AD dementia patients (Figure 1 and Figure 2). Lower total dietary score (i.e., lower adherence) and lower alcohol scores (i.e., higher intake) were associated with poorer memory performance (sβ 0.16 (0.00; 0.02); sβ 0.14 (0.00; 0.10)). A lower vegetable score (i.e., lower intake) was associated with poorer global cognition (sβ 0.20 (0.01; 0.08)) and poorer executive functioning (sβ 0.20 (0.01; 0.11)). A lower trans fat score (i.e., higher intake) was associated with poorer executive functioning as well (sβ 0.20 (0.00; 0.07)). There were no significant associations in MCI or controls.

## 4. Discussion

The main finding of this study is that in a memory clinic population, lower adherence to dietary guidelines was associated with poorer cognitive functioning. This association was mostly attributable to a lower vegetable intake and was most pronounced in AD dementia. In this group, suboptimal adherence to dietary guidelines was associated with more severe impairment in a range of cognitive domains. Previous literature in community-based samples showed that a healthy diet seems to reduce the risk of AD dementia [9,10,11,12,13], emphasizing a putative role for nutrition in the context of the prevention of dementia. Our study highlights the relevance of a healthy diet in the symptomatic stages of the disease, as associations between nutrition and cognition were most pronounced in AD dementia. This contrasts with the limited literature on this topic in clinical populations, where associations are more prominent in predementia stages. Adherence to the Mediterranean diet has been associated with global cognition in healthy controls, but not in MCI or AD dementia patients [14], and in a previous study, we found an association between vegetable intake and global cognition in cognitively healthy elderly [7]. While we did not specifically study the Mediterranean diet, the Dutch dietary guidelines are in line with the Mediterranean diet for the components vegetables, fruit, fibers, fish, salt, and trans fat. Our study extends previous studies in populations with cognitive impairment, which used a brief cognitive screener, such as the MMSE, by employing an extensive neuropsychological test battery assessing multiple cognitive domains. Based on our results, higher adherence to healthy dietary guidelines seems most relevant for the cognitive domains of memory and executive functioning. Both memory and executive functioning are core features of AD, which fits with the idea that impaired nutritional intake is closely related to the disease.

There are several explanations for the observed effects of nutrition and cognition in AD dementia. The association could be a consequence of the disease. As the disease progresses, patients might have difficulties in cooking healthy meals, forget to eat, or prefer unhealthy foods, resulting in poorer diet quality. In our sample, however, we found no difference in dietary scores between groups, suggesting that dietary intake is not changed in MCI or AD dementia compared to controls. Alternatively, a poor diet quality can lead to poorer cognition by offering less nutrients and energy to the brain. Moreover, an unhealthy diet could have a larger effect in a brain that already suffers from neurodegeneration, explaining our most prominent findings in AD dementia. It is conceivable that suboptimal vegetable intake leads to suboptimal levels of micronutrients, such as vitamins, minerals and fibers, which in turn negatively affect cognition [6]. For instance, intakes of folate, phylloquinone, and lutein, present in certain vegetables, have been associated with cognitive decline [5]. High intake of trans fat and alcohol might contribute to cardiovascular damage, which is particularly associated with executive dysfunctioning and is an important risk factor for more rapid cognitive decline in AD dementia [35]. Although our cross-sectional design precludes inferences on causality, intervention studies with combinations of nutrients have suggested that supplementation could indeed have beneficial effects on the brain [36,37,38].

Among the strengths of this study is the structured assessment of dietary quality through a validated questionnaire in a large clinical cohort of participants with different stages of cognitive impairment. The DHD-FFQ allowed analyses of total dietary quality, as well as specific dietary components, which are easy to translate into future recommendations for a memory clinic population. Furthermore, the use of an extensive and standardized neuropsychological test battery enabled detailed investigation of different cognitive domains. Among the potential limitations is the fact that the DHD-FFQ is a rather crude instrument. Covering 65% of daily dietary intake, it has been developed to rank participants in their adherence to the dietary guidelines. To limit estimation errors, household measures were asked and, when needed, caregivers completed the questionnaire for MCI and AD patients. Additionally, DHD-FFQ scores represent a monthly time frame, so any seasonal variability or sudden dietary changes are not accounted for. However, DHD-FFQ is a short and feasible instrument that can easily be administered in a memory clinic setting. We included BMI, but had no information on physical activity levels, so we cannot exclude the idea that some individuals score higher on dietary quality simply due to their higher energy intake.

Since we focused on a clinically relevant sample of patients with AD dementia, MCI, and controls, our findings have important clinical implications. We found that adherence to healthy dietary guidelines, especially regarding vegetable, trans fat, and alcohol intake, is of great importance, particularly in patients with mild AD dementia. As diet is a self-regulating and relatively easy target to prevent cognitive decline, patients visiting the memory clinic could benefit from adhering to healthy dietary guidelines. Further research is recommended to assess associations between dietary quality and cognitive performance, including controls without SCD. Longitudinal data and nutritional intervention studies are needed to support evidence on the importance of healthy dietary guidelines in a memory clinic population to prevent cognitive decline.

## Figures and Tables

**Figure 1 nutrients-12-00703-f001:**
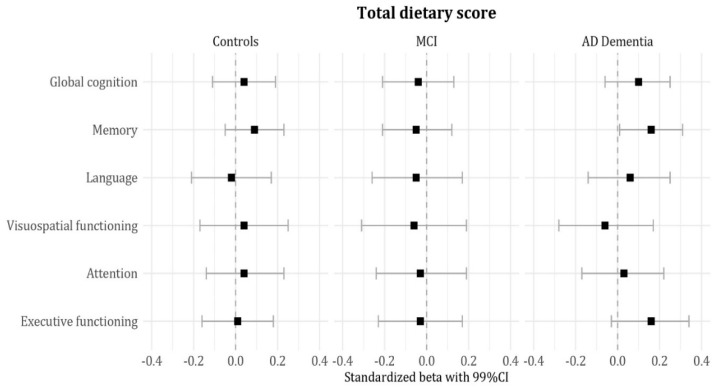
Associations between the total dietary score and cognitive performance by diagnosis group. Data are presented as standardized beta coefficients and (99%) confidence interval; effect sizes of age, sex, education, and BMI adjusted linear regression analyses.

**Figure 2 nutrients-12-00703-f002:**
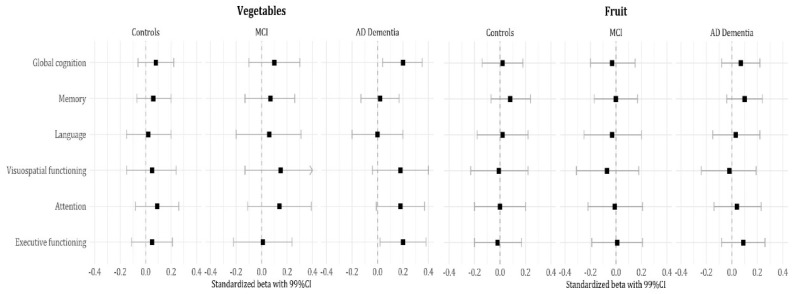
Associations between dietary component scores and cognitive performance by diagnosis group. Data are presented as standardized beta coefficients and (99%) confidence interval; effect sizes of age, sex, education, and BMI adjusted linear regression analyses.

**Table 1 nutrients-12-00703-t001:** Characteristics, dietary quality, and cognitive performance of the total study population and by diagnosis group.

Characteristics	Total Study Population	Controls	MCI	AD Dementia
*n*	357	133	90	134
Age (years)	65 ± 8.3	61 ± 7.2	66 ± 7.5 ^†^	68 ± 8.3 ^†^
Sex, female	165 (46)	62 (47)	34 (38)	69 (52)
Level of education	5 ± 1.2	6 ± 1.2	5 ± 1.2	5 ± 1.2 ^†^
BMI (kg/m^2^)	26 ± 4.1	27 ± 4.7	26 ± 3.4	25 ± 3.8 ^†^
Hypertension, yes	87 (24)	24 (18)	23 (26)	40 (30)
Hypercholesterolaemia, yes	50 (14)	14 (11)	16 (18)	20 (15)
Diabetes mellitus, yes	33 (9)	9 (7)	13 (14)	11 (8)
MMSE score	26 ± 3.3	28 ± 1.7	26 ± 2.5 ^†^	23 ± 3.0 ^†,‡^
**Dietary quality score ^a^**				
Total diet	53.9 ± 0.6	54.7 ± 11.6	53.0 ± 12.2	53.6 ± 11.0
Vegetables	6.1 ± 0.1	6.3 ± 3.0	6.5 ± 2.5	5.6 ± 2.7
Fruit	7.4 ± 0.2	7.6 ± 3.0	7.3 ± 3.3	7.1 ± 3.3
Fibers	7.2 ± 0.1	7.1 ± 2.1	7.4 ± 2.0	7.0 ± 2.0
Fish	5.6 ± 0.2	5.7 ± 3.5	5.8 ± 3.5	5.2 ± 3.2
Saturated fat	5.3 ± 0.2	5.4 ± 4.1	4.5 ± 4.1	5.8 ± 4.1
Trans fat	7.8 ± 0.2	7.8 ± 4.1	7.3 ± 4.4	8.0 ± 4.0
Salt	5.9 ± 0.2	6.0 ± 3.0	5.3 ± 3.0	6.1 ± 3.1
Alcohol	8.8 ± 0.1	8.8 ± 2.3	8.8 ± 2.4	8.8 ± 2.5
**Cognitive performance ^a^**				
Global cognition	-	0.5 ± 0.4	0.0 ± 0.3 ^†^	−0.6 ± 0.6 ^†,‡^
Memory	-	0.8 ± 0.6	−0.1 ± 0.6 ^†^	−0.7 ± 0.5 ^†,‡^
Language	-	0.5 ± 0.6	0.1 ± 0.4 ^†^	−0.5 ± 0.8 ^†,‡^
Visuospatial functioning	-	0.3 ± 0.3	0.2 ± 0.4	−0.4 ± 1.1 ^†,‡^
Attention	-	0.5 ± 0.6	0.0 ± 0.6 ^†^	−0.5 ± 0.8 ^†,‡^
Executive functioning	-	0.5 ± 0.6	0.0 ± 0.5 ^†^	−0.6 ± 0.8 ^†,‡^

Data are presented as mean ± SD; *n* (%); data for dietary quality are presented at a 0–80 scale for total dietary score and a 0–10 scale for dietary component scores; data for cognitive performance are presented as z-scores; raw test scores were converted into z-scores using the mean and SD of total study population (by definition total population: z = 0); MCI = mild cognitive impairment; AD = Alzheimer’s disease; BMI = Body Mass Index; MMSE = Mini-Mental State Examination; ^a^ adjusted for age, sex, education, and BMI upon post-hoc testing; ^†^ significantly different from the controls upon post-hoc testing; ^‡^ significantly different from MCI upon post-hoc testing.

**Table 2 nutrients-12-00703-t002:** Associations between dietary quality and cognitive performance in the total study population of controls, MCI and AD dementia patients.

Dietary Quality Score	Global Cognition	Memory	Language	Visuospatial Functioning	Attention	Executive Functioning
Total diet	0.06 (−0.00; 0.01)	0.10 (−0.00; 0.02)	0.01 (−0.01; 0.01)	−0.01 (−0.01; 0.01)	0.03 (−0.01; 0.01)	0.07 (−0.00; 0.01)
Vegetables	0.18 * (0.01; 0.07)	0.09 (−0.01; 0.07)	0.06 (−0.02; 0.05)	0.15 * (0.00; 0.08)	0.17 * (0.01; 0.09)	0.13 * (0.00; 0.08)
Fruit	0.09 (−0.01; 0.04)	0.13 (−0.00; 0.07)	0.05 (−0.02; 0.04)	0.01 (−0.03; 0.04)	0.05 (−0.02; 0.05)	0.07 (−0.01; 0.05)
Fibers	0.07 (−0.02; 0.07)	0.02 (−0.05; 0.06)	−0.02 (−0.06; 0.04)	0.09 (−0.02; 0.09)	0.06 (−0.03; 0.08)	0.07 (−0.02; 0.08)
Fish	0.04 (−0.02; 0.03)	0.07 (−0.02; 0.05)	−0.00 (−0.03; 0.03)	−0.01 (−0.04; 0.03)	0.04 (−0.02; 0.04)	0.03 (−0.03; 0.04)
Saturated fat	−0.09 (−0.03; 0.01)	−0.03 (−0.03; 0.02)	−0.08 (−0.04; 0.01)	−0.10 (−0.05; 0.01)	−0.08 (−0.04; 0.01)	−0.05 (−0.04; 0.02)
Trans fat	0.04 (−0.01; 0.03)	0.01 (−0.03; 0.03)	0.05 (−0.02; 0.03)	0.02 (−0.02; 0.03)	0.02 (−0.02; 0.03)	0.08 (−0.01; 0.04)
Salt	−0.05 (−0.04; 0.02)	0.04 (−0.03; 0.05)	0.02 (−0.03; 0.04)	−0.11 (−0.07; 0.01)	−0.06 (−0.05; 0.02)	−0.06 (−0.05; 0.02)
Alcohol	−0.03 (−0.04; 0.03)	0.09 (−0.01; 0.08)	−0.01 (−0.05; 0.04)	−0.03 (−0.06; 0.04)	−0.06 (−0.07; 0.02)	−0.03 (−0.05; 0.03)

Data are presented as standardized beta coefficients and (99%) confidence interval; effect estimates of age, sex, education, and BMI adjusted linear regression analyses; * *p* < 0.01.

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
