# Peer review of "A Suboptimal Diet Is Associated with Poorer Cognition: The NUDAD Project"

_nutrients, 2020, doi:10.3390/nu12030703_

Round 1

Reviewer 1 Report

Fieldhouse et al have cross-sectionally analysed the adherence to the Dutch guidelines for a healthy diet and the association to cognitive performance, both globally and individual domains, with vegetable intake showing the strongest associations.

There are a couple of major issues noted –

It is well established that neuropathology of AD develops up to 20 years before symptoms are clinically significant. Therefore, the current state of thought is that the most viable solution is to use this presymptomatic group for intervention studies to intervene in time to reduce AD risk or prevent it entirely. Your thoughts that diet modification once people have symptoms already is an option to prevent further decline is one not shared in the scientific community and is a concern to the use of the current manuscript.

The FFQ requires estimations of food intake over the previous month, there is potential therefore for misclassification due to limited accuracy in estimations, particularly with regard to MCI and AD participants. Consequently, it is preferable to use only data collected from individuals classified as healthy control at baseline. Results from participants with MCI, and especially AD, must be used with caution and this must be discussed as a limitation in the discussion.

The DHD-FFQ assesses intake over the past month and the authors then use this data to analyse associations with cognition. Generally, FFQs assess dietary intake over the past year, and this longer term of dietary intake is generally more reliable as a measure of usual dietary intake than just over the previous month, which has seasonal variability not accounted for. Were participants questioned as to whether they had significantly changed their diet in the past month compared to the past year, for example cut out a food item like meat, gluten etc. Again, this issue needs to be addressed in the discussion as could affect the interpretation of the findings.

With longitudinal data available, why is a longitudinal analysis not included which will make the paper of much higher scientific quality than just cross-sectional

Minor changes suggested -

The abstract, lines 34 – 35, state associations were observed but not the direction of these associations which is required to be included.

For the reader without specific dementia knowledge, can the term MCI be expanded on to inform readers that it is characterised by cognitive dysfunction at a level that is not of sufficient severity for a diagnosis of AD, it does not significantly restrict activities of daily living, and often but not always precedes AD.

Subjective cognitive decline is mentioned on line 70 without further mention in the manuscript. The reporting of subjective memory complaints has been shown to be a significant predictor of future cognition and analysing the differences between those with and without subjection cognitive decline could add an interesting and novel element to the manuscript.

There are a few English language edits required, including

line 77. “performed normal” should be changed to “performed normally”

If the acronym MMSE is added on line 57 then the acronym alone can be used on line 70

Line 212 supplementation is incorrectly spelled

The final sentence of the paragraph line 225 – 226 is superfluous as is a repetition of lines 211 – 212

Information on the validation of the DHD-FFQ should be included in the methods section.

Lines 111 – 113 state the percentage of missing data. Can the authors confirm if this was missing data due to the participants not completing that test, or whether it is missing as the participants were unable to complete the test due to cognitive decline, in which case where these results marked as a zero and not missing values?

Line 192 states no other previously published manuscripts have used a cognitive battery but only used a brief screener. Whilst this may be true for the Dutch guidelines for a healthy diet, it certainly is not true for MeDi, which is included in the previous sentence as in line with the Dutch guidelines. A review in 2017 by Loughrey et al. in Advances in Nutrition, analyses 15 manuscripts of which 7 of these included several cognitive domains and not only global cognition in their analysis. This sentence either needs stating authors are only referring to the analysis of manuscripts directly analysing the Dutch guidelines (not what the reviewer recommends) or needs editing to show the true MeDi extant literature. The same applies to lines 56 – 58.

Referencing needs correcting, reference numbers should be placed before the punctuation.

Author Response

Author’s Reply to Review Report - Reviewer 1

Fieldhouse et al have cross-sectionally analyzed the adherence to the Dutch guidelines for a healthy diet and the association to cognitive performance, both globally and individual domains, with vegetable intake showing the strongest associations.

We would like to thank the reviewer for the comments and suggestions, which all contributed to improvement of the manuscript. In point-by-point fashion we provide an overview of the changes that were made.

- Major issues:

Point 1: It is well established that neuropathology of AD develops up to 20 years before symptoms are clinically significant. Therefore, the current state of thought is that the most viable solution is to use this presymptomatic group for intervention studies to intervene in time to reduce AD risk or prevent it entirely. Your thoughts that diet modification once people have symptoms already is an option to prevent further decline is one not shared in the scientific community and is a concern to the use of the current manuscript.

Response 1: The reviewer rightfully pointed that the neuropathology of AD has been shown to develop long before clinical symptoms arise, and subsequently, research regarding the prevention of AD has shifted to presymptomatic stages. In healthy populations, the association between nutrition and cognition is already widely recognized, as specific food groups and diets have been shown to prevent cognitive decline. However, there is a lack of literature on the association between nutrition and cognition in a memory clinic population. For this specific population, prevention is too late and a therapy is not yet within reach. Visitors of a memory clinic often remain with a much reported question of what can be done in this stage of the disease. To answer this clinically relevant question, this study takes a first step in investigating the association between nutrition and cognition in a specific memory clinic population.

Point 2: The FFQ requires estimations of food intake over the previous month, there is potential therefore for misclassification due to limited accuracy in estimations, particularly with regard to MCI and AD participants. Consequently, it is preferable to use only data collected from individuals classified as healthy control at baseline. Results from participants with MCI, and especially AD, must be used with caution and this must be discussed as a limitation in the discussion.

Response 2: The
Dutch Healthy Diet - Food Frequency Questionnaire (DHD-FFQ) assesses frequencies of food intake, the participants were asked to estimate portion sizes of particular food groups. To limit misclassification of these estimations everyday household measures are used. When AD or MCI patients were not able to fill in the questionnaire, the DHD-FFQ was completed by the caregiver. We acknowledge we did not report this well in the manuscript, it has been added to the Materials and Methods section. Additionally, potential estimation errors have been added as a limitation in the Discussion section.

(page 2-3, line 95-96 in the revised manuscript)
“When a patient was not able to fill in the questionnaire, the DHD-FFQ was completed by a caregiver.”
(page 7, line 225-226 in the revised manuscript)
“To limit estimation errors household measures were asked, and when needed, caregivers completed the questionnaire for MCI and AD patients.”

Point 3: The DHD-FFQ assesses intake over the past month and the authors then use this data to analyze associations with cognition. Generally, FFQs assess dietary intake over the past year, and this longer term of dietary intake is generally more reliable as a measure of usual dietary intake than just over the previous month, which has seasonal variability not accounted for. Were participants questioned as to whether they had significantly changed their diet in the past month compared to the past year, for example cut out a food item like meat, gluten etc. Again, this issue needs to be addressed in the discussion as could affect the interpretation of the findings.

Response 3: The fact that we did not account for potential seasonal variability and/or sudden dietary changes is added as a limitation in the Discussion section.

(page 7, line 229-231 in the revised manuscript)
“Additionally, DHD-FFQ scores represent a monthly time frame, any seasonal variability or sudden dietary changes are not accounted for.”

Point 4: With longitudinal data available, why is a longitudinal analysis not included which will make the paper of much higher scientific quality than just cross-sectional.

Response 4: We agree that longitudinal analysis will definitely improve the quality of our manuscript. Unfortunately, we are currently in the process of collecting longitudinal data of this cohort, to avoid confusion we slightly adjusted the description of our current project in the Materials and Methods section.

(page 2, line 66-67 in the revised manuscript)
The ‘Nutrition the Unrecognized Determinant for Alzheimer’s Disease’ (NUDAD) project prospectively studies nutritional determinants in AD and pre-dementia stages (deleted: with three year clinical follow-up).”

-
 Minor changes:

Point 5: The abstract, lines 34 – 35, state associations were observed but not the direction of these associations which is required to be included.

Response 5: This sentence has been adjusted in the abstract.

(page 1, line 33-36 in the revised manuscript)
“In the total population, linear regression analyses showed lower vegetable intake is associated with poorer global cognition, visuospatial functioning, attention and executive functioning.”

Point 6: For the reader without specific dementia knowledge, can the term MCI be expanded on to inform readers that it is characterised by cognitive dysfunction at a level that is not of sufficient severity for a diagnosis of AD, it does not significantly restrict activities of daily living, and often but not always precedes AD.

Response 6: Characterization of the term MCI, including a reference of MCI criteria, have been added to the Materials and Methods section.

(page 2, line 76-78 in the revised manuscript)
“MCI is an umbrella term characterizing cognitive impairment beyond what is expected for age and education, but not affecting activities of daily living and not as severe to fulfill criteria for dementia[20].”

Point 7: Subjective cognitive decline is mentioned on line 70 without further mention in the manuscript. The reporting of subjective memory complaints has been shown to be a significant predictor of future cognition and analyzing the differences between those with and without subjection cognitive decline could add an interesting and novel element to the manuscript.

Response 7: As the reviewer rightfully noted, subjects with subjective cognitive decline (SCD) are an interesting population. Our SCD subjects all scored normal on extensive neuropsychological testing, so within this particular memory clinic population we consider them as controls, while we acknowledge that they visit the memory clinic because of cognitive complaints. Please note that the current study is focused a specific population with memory complaints. Indeed, including a subgroup without SCD would be interesting and novel, we have added this to the Discussion section.

(page 8, line 240-241 in the revised manuscript)
Further research is recommended to assess associations between dietary quality and cognitive performance, including controls without SCD.”

Point 8: There are a few English language edits required, including:
- Line 77. “performed normal” should be changed to “performed normally”
(page 2, line 79 in the revised manuscript)

- If the acronym MMSE is added on line 57 then the acronym alone can be used on line 70 (page 2, line 57 and 70 in the revised manuscript)

- Line 212 supplementation is incorrectly spelled
(page 7, line 219 in the revised manuscript)

- The final sentence of the paragraph line 225 – 226 is superfluous as is a repetition of lines 211 – 212
(page 8, line 234 in the revised manuscript)

Response 8: We applied the suggested edits regarding English language.

Point 9: Information on the validation of the DHD-FFQ should be included in the methods section.

Response 9: We added the validation study to the Materials and Methods section.

(page 2, line 88-89 in the revised manuscript)
“The validated Dutch Healthy Diet - Food Frequency Questionnaire (DHD-FFQ) was used to assess averaged daily dietary quality in the previous month [22, 23].”

Point 10: Lines 111 – 113 state the percentage of missing data. Can the authors confirm if this was missing data due to the participants not completing that test, or whether it is missing as the participants were unable to complete the test due to cognitive decline, in which case where these results marked as a zero and not missing values?

Response 10: The cognitive data was collected during a standardized neuropsychological test battery of one hour. Missing data in this assessment is not entirely random, since most severely impaired patients might experience more difficulties in finishing the test battery in this fixed time period. We can, therefore, not confirm missings are either due to participants not completing the test or whether they were not able to perform the test. All missing data were marked as missing values.

Point 11: Line 192 states no other previously published manuscripts have used a cognitive battery but only used a brief screener. Whilst this may be true for the Dutch guidelines for a healthy diet, it certainly is not true for MeDi, which is included in the previous sentence as in line with the Dutch guidelines. A review in 2017 by Loughrey et al. in Advances in Nutrition, analyses 15 manuscripts of which 7 of these included several cognitive domains and not only global cognition in their analysis. This sentence either needs stating authors are only referring to the analysis of manuscripts directly analysing the Dutch guidelines (not what the reviewer recommends) or needs editing to show the true MeDi extant literature. The same applies to lines 56 – 58.

Response 11: We agree that there are multiple studies assessing associations between the Mediterranean diet and cognitive domains within healthy older adults, like the suggested review by Loughrey et al. However, our focus is on memory clinical population and within this specific population there is a lack of knowledge about nutrition and cognitive performance on an extensive test battery. We have added the suggested reference to the literature regarding healthy populations in the Introduction and Discussion sections. Furthermore, in the sentence stated by the reviewer we emphasized our focus on a population with cognitive impairment.

(page 2, line 47-52 in the revised manuscript)
“In cognitively normal populations, food groups including vegetables, fruit or fish[3-8] and healthy dietary patterns such as the Mediterranean diet, dietary approaches to stop hypertension (DASH) or Mediterranean-DASH intervention for neurodegenerative delay (MIND)[9-13] have all been associated with less cognitive decline or reduced risk for mild cognitive impairment (MCI) or AD dementia.”

(page 7, line 187-189 in the revised manuscript)
“Previous literature in community based samples showed that a healthy diet seems to reduce the risk for AD dementia[9-13], emphasizing a putative role for nutrition in the context of prevention of dementia.”

(page 7, line 197-199 in the revised manuscript)
“Our study extends previous studies in populations with cognitive impairment, which used a brief cognitive screener such as the MMSE, by employing an extensive neuropsychological test battery assessing multiple cognitive domains.”

Point 12: Referencing needs correcting, reference numbers should be placed before the punctuation.

Response 12: We have corrected all reference numbers in the revised manuscript.

Reviewer 2 Report

This manuscript discusses an analysis of the data of the NUDAD project. The research question is interesting, but the presentation can be improved.

#59-62: I had hoped that the authors would use more of the available data in the paper. I am thinking of macronutrient intake; co-morbidities etc. If the authors want to publish those data at a later stage, they well could be confronted with the present data.

#74: You did not report the lab tests. 

#124: You chose to set the p<0.01 as significant. I appreciate this as the chance of finding false positives could be high. However, what I do not understand that still the data are represented as 95% CI. I think all data should be 99% CI and that would make your message much clearer.

#126: It is unclear whether SD or SE is used for data in the text or table 1.

184: I would be more specific here. You only tested certain aspects of nutrition. 

#247-253: It seems that members of the project team are industry-paid members. How is the COI for the present report? Is there no funding from the industry? 

Author Response

Author’s Reply to Review Report - Reviewer 2

This manuscript discusses an analysis of the data of the NUDAD project. The research question is interesting, but the presentation can be improved.

We would like to thank the reviewer for the encouraging words and suggestions for improvement of the manuscript.

Point 1: #59-62: I had hoped that the authors would use more of the available data in the paper. I am thinking of macronutrient intake; co-morbidities etc. If the authors want to publish those data at a later stage, they well could be confronted with the present data.

Response 1: Thank you for your comment and interesting question. It would indeed be interesting to assess detailed associations between other nutritional factors and cognitive performance. In this study, we specifically focus on the associations of diet quality with cognitive performance. We have, however, added comorbidity numbers to the descriptives of our cohort in Table 1.

Point 2: #74: You did not report the lab tests. 

Response 2: Laboratory tests were performed as part of the cognitive work-up, to rule out somatic causes for cognitive impairment, and to establish a clinical diagnosis. In our opinion reporting these laboratory tests will not have additional value for the research goals of this study.

Point 3: #124: You chose to set the p<0.01 as significant. I appreciate this as the chance of finding false positives could be high. However, what I do not understand that still the data are represented as 95% CI. I think all data should be 99% CI and that would make your message much clearer.

Response 3: This is a valid point. We have changed all data in text, table 2 and figures to 99% confidence interval (CI). Additionally, we have marked significant associations (<0.01) with an asterisk and a bold font in table 2.

Point 4: #126: It is unclear whether SD or SE is used for data in the text or table 1.

Response 4: We are sorry for the inconvenience. We corrected this issue in the revised manuscript, all data in text and table 1 are now presented with correct standard deviations (SD).

Point 5: #184: I would be more specific here. You only tested certain aspects of nutrition. 

Response 5: We rephrased this sentence in the Discussion section.

(page 7, line 189-191 in the revised manuscript)
Our study highlights the relevance of vegetable intake in the symptomatic stages of the disease, as associations between nutrition and cognition were most pronounced in AD dementia.”

Point 6: #247-253: It seems that members of the project team are industry-paid members. How is the COI for the present report? Is there no funding from the industry? 

Response 6: The NUDAD project is partly funded by the industry, however the authors of this manuscript are not working for these industries. Furthermore, the industrial partners had no role or contribution in the study, as described in the COI of the manuscript.

(page 8, line 265-267 in the revised manuscript)
Conflicts of Interest: The authors declare no conflict of interest. The funders had no role in the design of the study; in the collection, analyses, or interpretation of data; in the writing of the manuscript, or in the decision to publish the results.”

Round 2

Reviewer 1 Report

Thank you to the authors for the response to the previous comments. They have all be answered satisfactorily and the additions have made this a higher quality presentation.